# The Different Composition of Coumarins and Antibacterial Activity of *Phlojodicarpus sibiricus* and *Phlojodicarpus villosus* Root Extracts

**DOI:** 10.3390/plants13050601

**Published:** 2024-02-22

**Authors:** Maria T. Khandy, Valeria P. Grigorchuk, Anastasia K. Sofronova, Tatiana Y. Gorpenchenko

**Affiliations:** 1Laboratory of Cell and Developmental Biology, Federal Scientific Center of East-Asia Terrestrial Biodiversity, Far Eastern Branch of the Russian Academy of Sciences, 159 Stoletiya Street, Vladivostok 690022, Russiagorpenchenko@biosoil.ru (T.Y.G.); 2Department of Medical Biology and Biotechnology, School of Medicine and Life Sciences, Far Eastern Federal University, FEFU Campus, 10 Ajax Bay, Russky Island, Vladivostok 690922, Russia

**Keywords:** pyranocoumarins, furacoumarins, hydroxycoumarins, coumarins, HPLC-MS, *Phlojodicarpus sibiricus*, *Phlojodicarpus villosus*, *Apiaceae*, secondary metabolites, phenolic compounds, antibacterial activity

## Abstract

*Phlojodicarpus sibiricus*, a valuable endangered medicinal plant, is a source of angular pyranocoumarins used in pharmacology. Due to limited resource availability, other pyranocoumarin sources are needed. In the present research, the chemical composition of a closely related species, *Phlojodicarpus villosus*, was studied, along with *P. sibiricus*. High-performance liquid chromatography and mass-spectrometric analyses, followed by antibacterial activity studies of root extracts from both species, were performed. *P. sibiricus* and *P. villosus* differed significantly in coumarin composition. Pyranocoumarins predominated in *P. sibiricus*, while furanocoumarins predominated in *P. villosus*. Osthenol, the precursor of angular pyrano- and furanocoumarins, was detected in both *P. sibiricus* and *P. villosus*. Angular forms of coumarins were detected in both species according to the mass-spectrometric behavior of the reference. Thus, *P. villosus* cannot be an additional source of pyranocoumarins because their content in the plant is critically low. At the same time, the plant contained large amounts of hydroxycoumarins and furanocoumarins. The extracts exhibited moderate antibacterial activity against five standard strains. The *P. villosus* extract additionally suppressed the growth of the Gram-negative bacterium *E. coli*. Thus, both *Phlojodicarpus* species are promising for further investigation in the field of pharmaceuticals as producers of different coumarins.

## 1. Introduction

Coumarins are a large group of secondary plant metabolites. Their structural diversity provides different biological activities within the group of these substances, including antioxidant [1], antibacterial [2,3], antiviral [4,5], anticoagulant [6], anti-inflammatory [7], and antitumor [8,9,10]. In recent years, studies of the model plant species *Nicotiana* and *Arabidopsis* have significantly expanded our understanding of the biosynthesis, accumulation, secretion, and chemical modification of simple coumarins and their modes of action against plant pathogens [11]. Complex coumarins are divided into three main groups: furanocoumarins, pyranocoumarins, and benzocoumarins. The chemical diversity and pharmacological activities of furanocoumarins and benzocoumarins are widely represented in the literature due to their greater availability as they are synthesized in many plant species [12]. Pyranocoumarins are a rare group of coumarins that are distributed exclusively between two families, *Apiaceae* and *Rutaceae* [13]. A wide range of biological activities and the use of coumarin-containing drugs in the clinic have contributed to the growing interest in this class of heterocycles. This group of rare coumarins remains a blank spot in science.

Similar to all coumarins, furano- and pyranocoumarins have a wide range of biological activities and are widely used not only in ethnopharmacology but also in modern medicine. In most cases, the biological activity of coumarins is associated with the structural arrangement of the main pyran ring. Furanocoumarins and benzocoumarins are divided into linear and angular according to the different positions of the main ring [13,14], while pyranocoumarins can be linear (6,7-position), angular (5,6-; 7,8-position) [13,14,15], and condensed (4,5-position) [16].

The main source of 7,8-angular pyranocoumarins is *Phlojodicarpus sibiricus* (Fisch. Ex Spreng.) Koso-Pol., a rare and valuable medicinal plant found in Russia and Mongolia [17]. According to various authors, *P. sibiricus* contains umbelliferon [18], khellactone, 4′-*O*-methyl ether of khellactone, hyuganin D, suxdorphin [19], visnadine [20], and dihydrosamidine [21]. Olennikov and coauthors, using high-performance liquid chromatography with a diode array detector and a triple quadrupole detector with ionizing electrospray, determined the composition of coumarins, flavonoids, and phenylpropanoids in the roots of *P. sibiricus*. A total of 41 compounds were identified [22]. Extensive research has been conducted to determine the biological activity of these substances in extracts and tinctures of *P. sibiricus* in relation to the nervous and cardiovascular systems of rats, sick people, dorsal ganglia explants of chicken embryos, the 3T3-L1 preadipocyte cell line, and red blood cell donor blood [22,23,24,25,26,27]. Vasodilator drugs were obtained based on pyranocoumarins such as visnadine and dihydrosamidine isolated from the extracts of *P. sibiricus*. There are no scientific data on the effect of its extracts on the growth and development of bacteria, even though such an antibacterial effect has been demonstrated for the pyranocoumarins of other plant species [28]. A limited number of linear pyranocoumarins studied demonstrated low antibacterial activity compared to other coumarins [28,29]. Despite this, substitutions at different positions of the pyran ring and the addition of different functional groups can significantly enhance the antibacterial effect [28,29,30]. Over the past decade, various coumarin-based antibiotic hybrids from other plant species have been developed, and most of them have demonstrated potential antibacterial activity [30]. Thus, most types of biological activity inherent in coumarins (anticoagulant, antidiabetic, and neuroprotective), with the exception of antibacterial activity, have been shown for angular pyranocoumarins. The assessment of the antimicrobial activity of the extracts in this work may increase our understanding of the compounds contained in these extracts.

*P. sibiricus* is a rare plant with a limited growing area. The widespread use of this species has led to a shortage of plant materials. A solution to this challenge may be the use of closely related species for the production of drugs based on 7,8-angular pyranocoumarins. Investigations of the secondary metabolite diversity of closely related species can serve as the basis for discovering new rare coumarins and studying their properties.

Plants of the genus *Phlojodicarpus* belong to the family *Apiaceae* and are represented by only three species: *P. sibiricus*, *P. villosus* (Turcz. Ex Fisch. et C. A. Mey.) Lebed, and *P. komarovii* Gorovoj. *P. komarovii* is a rare endemic of Russia, the description of which is not confirmed by data from the past 50 years [17]. *P. villosus* is distributed in the Arctic regions of the north-east of Russia [17,31]. Outside Russia, it is found in Mongolia [17]. Available research works on the phytochemical composition of *P. villosus* contain information on several coumarin derivatives: dihydrosamidine [21], visnadine [20], decursin, and agasyllin with high concentrations [32]. No modern data on secondary metabolites could be found. Thus, *P. villosus* is a promising subject for scientific research and a producer of rare types of coumarins. To obtain a more complete and informative picture, it is necessary to analyze this species using modern methods for isolating and determining its chemical components.

Thus, the goals of this work are to study the qualitative composition of root extracts from two species of *P. sibiricus* and *P. villosus* for alternative sources of a rare group of coumarins (pyranocoumarins) and to investigate the antibacterial activity of the obtained extracts.

## 2. Results

### 2.1. Coumarin Composition

The HPLC-UV-HR-MS/MS^2^ (high-performance liquid chromatography–high-resolution tandem mass spectrometry) method was used to determine the coumarin compounds present in the crude methanolic extracts of the *P. sibiricus* roots from two different regions and *P. villosus*.

More than fifty peaks (Figure 1) were tentatively assigned to various coumarin derivatives due to their well-defined UV/VIS-DAD spectra with an absorption maximum at 320–330 nm. The recognition of all defined compounds was based on their chromatographic behavior, UV and mass-spectral data, and comparison with reference compounds and scientific publications [22,33,34,35,36,37,38].

While studying the coumarin composition and mass-spectrometric characteristics of the extract components, we found a noticeable difference in fragmentation patterns between components of with the same masses from the extracts of *P. sibiricus* and *P. villosus*. These turned out to be different groups of coumarins. As was previously published [22], *P. sibiricus* roots and herb were excessively rich in angular pyranocoumarins—khellactone derivatives. Our assumption that *P. villosus* roots contained another group of coumarins needed to be checked.

To begin with, two available angular-type coumarin standards were analyzed, UV and MS data were obtained, and their mass-spectrometric behavior was studied and compared. The pyranocoumarin khellactone diacyl ester visnadine (Sigma, St. Louis, MI, USA) and the furanocoumarin vaginidiol diacyl ester peucenidin (previously isolated from *Ferulopsis hystrix* [34]) were acquired. Figure 2 and Figure 3 show the structures and MS/MS^2^ spectra of the studied coumarins. It should be noted that all MS experiments were carried out with the simultaneous registration of negative and positive ions, and positive ion detection turned out to be more informative in the recognition of disubstituted coumarin derivatives. So, adduct ions with compositions [M+Na]^+^ and [M+NH_4_]^+^ showed the most intensive signals; also, dimeric sodium adduct ions [2M+Na]^+^ were easily observed in the spectra of both studied compounds, whereas protonated ions [M+H]^+^ were completely absent in both cases. This was where the similarities between these two standard compounds ended. Visnadine showed a signal with dominant intensity at *m/z* 329 due to the neutral loss of acetic acid C_2_H_4_O_2_ (the RCOOH unit from C-4′; Figure 2A,B). These characteristic fragment ions could be perfectly observed both in the full scan spectrum and in the MS^2^ fragmentation spectrum of precursor ions [M+NH_4_]^+^ (Figure 2A,D). In addition, sodium adduct ions [M+Na]^+^ fragmentation demonstrated a neutral loss of C_2_H_4_O_2_: ions [M+Na-C_2_H_4_O_2_]^+^ at *m/z* 351. Another fragment ions signal at *m/z* 245 (Figure 2A–D) was formed by the elimination of two units: acid C_2_H_4_O_2_ and ketene C_5_H_8_O (the RCO unit from C-3′; Figure 2B). Summarizing the above, all necessary information for the structural identification of visnadine could be found directly in the full scan spectrum. The main diagnostic ions with the composition [M+H-RCOOH]^+^ corresponded to the elimination of an acid molecule from position 4′ (R1 on the schematic structure of pyranocoumarins; see Figure 1) with the formation of a double bond between atoms 3′ and 4′. Further, the 3′-O-acyl moiety (R2 on the schematic structure of pyranocoumarins; see Figure 1) could be calculated.

As for peucenidin, a completely different fragmentation pattern was displayed (Figure 3). The most intensive fragment ions at *m/z* 245 and 277 could be observed and formed due to successive losses of acid and ketene molecules (from positions 9 and 1′; Figure 3B) and two acids, respectively. The removal of both acids was more energetically beneficial because this would result in the formation of two double bonds rather than a triple, as in the case of visnadine. Unfortunately, the only signal informative for structural identification at *m/z* 327 (formed by the elimination of one acid molecule from position 9) demonstrated a very low intensity in the full scan spectrum (less than 1%) (Figure 3A). So we had to study the fragmentation spectra of available precursors (Figure 3C,D). It turned out that the target information could be obtained by measuring the MS^2^ spectrum of precursor [M+Na]^+^ ions. The fragment ions at *m*/*z* 349 (due to loss of the acid molecule from position 9; Figure 2B,C; R1 on the schematic structure of furanocoumarins; see Figure 1) had been detected. Moreover, the 1′-O-acyl moiety (R2 on the schematic structure of furanocoumarins; see Figure 1) could be calculated. We assumed that all of the above was also true for all defined pyranocoumarin and furanocoumarin diacyl esters in their entirety.

Thus, 37 diacyl esters of khellactone (21) and vaginidiol (16) were identified in crude aqueous methanol root extracts using the method described above. Adduct ions with compositions [M+Na]^+^ and [M+NH_4_]^+^ were observed as base peaks in the full scan spectra of all defined diacyl derivatives, which was typical for angular-type coumarins [33]. Also, six monoacyl derivatives were recognized, including two khellactone and four vaginidiol esters.

Coumarin glycosides were detected in every analyzed sample. As for the glycoside derivatives, their mass-spectrometric behavior was similar. The most abundant of all defined glycoconjugates, compound 2 was chosen as an example (Figure 1). In the ESI conditions, component 2 showed the formation of deprotonated ions [M-H]^−^ at *m*/*z* 455.119, negative adduct ions [M+HCOO]^−^ at *m*/*z* 501.123, and positive sodium adduct ions [M+Na]^+^ at *m*/*z* 479.114, which was in excellent agreement with the molecular formula C_20_H_24_O_12_. Also, a positive full scan spectrum of compound 2 demonstrated an intensive signal of fragment ions at *m*/*z* 325.085 ([M+H-C_5_H_8_O_4_]^+^) due to the loss of the pentoside moiety and a low signal at *m/z* 163.042 ([M+H-C_5_H_8_O_4_-C_6_H_10_O_5_]^+^) generated by the sequential removal of pentoside and hexoside moieties. Moreover, the MS^2^ fragmentation spectrum of precursor ions [M-H]^−^ displayed intensive signals at *m*/*z* 293.090 ([M-H-C_9_H_6_O_3_]^−^, formed by the elimination of the umbelliferone molecule) and at *m*/*z* 161.025 ([C_9_H_5_O_3_]^−^, due to complete deglycosylation). Summarizing the above, compound 2 was identified as umbelliferone-O-pentosyl-O-hexoside or apiosylskimmin, previously described in *P. sibiricus* [22]. Simultaneously, twelve di- (seven) and mono- (five) glycosylated coumarin derivatives were defined in root extracts of Phlojodicarpus [22,34,37].

Compound 17 was observed as the predominant signal of protonated ions at *m/z* 245.115 corresponding to the molecular formula C_15_H_16_O_3_. Also, its positive full scan spectrum showed the signal of fragment ions at *m/z* 189.059 ([M+H-C_4_H_8_]^+^) due to the loss of the prenil moiety. So, component 17 was identified as the well-known prenylated methoxycoumarin osthol [36].

All data important for identification are collected in Table 1. The compound numbers correspond to those shown in Figure 1.

The qualitative composition as well as the quantitative content of different groups of coumarins varied depending on the sample (Figure 4).

Pyranocoumarins were found to be the predominant group of coumarins in *P. sibiricus*, but their qualitative composition varied greatly between populations. In all, 26 different coumarins were found in samples of population I of *P. sibiricus*, while samples of population II of *P. sibiricus* were rich mainly in the content of the coumarins visnadine (30) and dihydrosamidine (31).

The predominant group of coumarins in *P. villosus* was furanocoumarins (71% of the total coumarin content). *P. villosus* was characterized by a high hydroxycoumarin content, accounting for 28% of the total coumarin content (Figure 4).

### 2.2. Antibacterial Activity

Optical density was measured for each half hour of incubation of the bacterial suspension cultures with *P. sibiricus* and *P. villosus* root extracts (25 mg biomass extract in 1 mL of the final solution). From these data, graphs depicting the growth dynamics of individual bacterial strains over 24 h were plotted. As can be seen in the graphs below, the extracts had different effects on the growth of Gram-negative and Gram-positive bacteria (Figure 5 and Figure 6). Both extracts suppressed the growth of *S. enterica* and *S. aureus*. At the same time, the *P. villosus* extract suppressed the growth of the Gram-negative bacterium *E. coli*. In other variants, no suppressive effect of the extracts against the tested bacteria was observed.

## 3. Discussion

The results of HPLC-MS analysis indicated that *P. sibiricus* and *P. villosus* differ significantly in the composition of coumarins; pyranocoumarins predominate in *P. sibiricus*, while furanocoumarins predominate in *P. villosus*. Based on the results of the fragmentation of furano- and pyranocoumarins from *P. sibiricus* and *P. villosus* in comparison with the standards of visnadine and peucenidin, only the angular forms of coumarins were found in both species. The differences in the mass spectra of linear and angular pyranocoumarins are confirmed by data from the literature [13,33].

Moreover, significant differences were found in the composition of *P. sibiricus* pyranocoumarins collected from two different areas. This pattern was previously shown for *P. sibiricus* by Olennikov’s research group, where the chromatographic profiles of four samples of *P. sibiricus* growing in the central region of Chita, the center of Yakutia, and the southern regions of Mongolia and Buryatia differ [22]. The authors hypothesized that the chemical composition of plant species growing in the Asian region differs due to different climatic conditions. Different compositions of metabolites were detected for *Angelica archangelica* L. [39] and *Crithmum maritimum* L. plants collected in different geographical areas [40]. Our study supports this hypothesis, as the composition of pyranocoumarins varies significantly among different geographic populations of *P. sibiricus*.

In plants, angular furano- and pyranocoumarins are derived from a common precursor (Figure 7). In the two species, the aglycones of furano- and pyranocoumarins are identical, but in each species, only one form of the substance predominates significantly: in *P. sibiricus* pyranocoumarins predominate, while furanocoumarins are more common in *P. villosus*. The distribution of pyranocoumarin forms within families producing these substances shows a significant predomination of one of the structural forms within one family [13]. Typically, the predominant metabolites are most characteristic of plants of the same family, related genera, and species. Both furano- and pyranocoumarins are found among the *Apiaceae* and *Rutaceae* families, but furanocoumarins are more widely distributed [12,13]. The angular furanocoumarins are mainly present in the genera *Heracleum*, *Pastinaca*, and *Pimpinella* and are almost absent from *Angelica*, *Peucedanum*, *Prangos*, and *Seseli*. The linear furanocoumarins are present in the genera *Prangos*, *Angelica*, and *Peucedanum* [12]. The presence of furanocoumarins in the genus *Phlojodicarpus* has not yet been demonstrated. Our data revealed that the quantitative ratio of pyrano- and furanocoumarins in the two species differs greatly—by 40 or more times (Figure 4). Climatic conditions likely exert a great influence on the diversity of individual metabolites within the group and their modifications (glycosylation, acylation), while the initial stages of the biosynthesis of the main group-determining enzymes (pyrano- or furanocoumarins) are controlled by the genotype.

Simple coumarins such as 4-hydroxycoumarin, umbelliferone, esculetin, scopoletin, etc., are widely distributed in different plant species. Therefore, the main pathways for the biosynthesis of these substances have already been well studied. Many enzymes involved in the biosynthesis of these substances have been isolated and characterized [47,48]. Some stages of the biosynthesis of simple coumarins have been successfully implemented in microbial systems expressed in *Escherichia coli* [47]. The Min-Juan Xu group, in a series of articles [49,50], reconstituted the biosynthetic pathways, including plant-derived umbelliferone prenyltransferases and the key enzymes XimD and XimE from *Streptomyces xiamenensis*, for the production of furano- and pyranocoumarins in microbial hosts. At the same time, by modeling various substrates and combinations of these enzymes, they were able to obtain only trace amounts of the angular forms of pyrano- and furanocoumarins [49,50]. Consensus is rather more complicated with the subsequent stages of the biosynthesis of complex coumarins. Some authors have suggested that the determination of furanocoumarin type is based on the prenylation position of the common precursor of all furanocoumarins, umbelliferone, at C6 or C8, which gives rise to the linear psoralen or angular angelicin derivatives, respectively [47,48,51]. Single prenyltransferase, such as PcPT, opens the pathway to linear furanocoumarins in parsley but may also catalyze the synthesis of osthenol, the first intermediate committed to the angular furanocoumarin pathway, in other plants [51]. Also, umbelliferone was exposed to a prenyl donor substrate (dimethylallyl pyrophosphate (DMAPP)) and transformed into demethylsuberosin or osthenol by prenyltransferase [47,48]. It was also shown that the biosynthesis of xanthyletin and seselin may be formed by a mevalonate-independent pathway from osthenol in *Thamnosma montana* [48,52]. Perhaps this stage is no less important for the formation of linear and angular pyranocoumarins.

It is interesting to note that osthol, a methylated derivative of osthenol, a precursor of angular pyrano- and furanocoumarins, was found in *P. sibiricus* and *P. villosus* (Figure 5). *P. villosus* has not been previously reported as a source of osthol. The investigation of crucial ferments directly opens the possibility of pyranocoumarin biosynthesis in similar genera and may clarify the biosynthetic mechanisms of complex coumarins in various plant systems.

The other important result of this work is the fairly high content of hydroxycoumarins in the studied species, *P. sibiricus* (13–17 mg/g) and *P. villosus* (31 mg/g) (Figure 4). For *P. villosus*, 30% of all coumarins are hydroxycoumarins, which makes this species a promising source of umbelliferone derivatives [10,13]. Various natural modifications of molecular structures determine the specific properties of the substances and, as a result, their medical use. Osthol, found in both species, is known as a calcium channel blocker [53]. Moreover, osthenol showed the most effective antibacterial activity against Gram-positive bacteria with minimum inhibitory concentration (MIC) values ranging between 125 and 62.5 μg/mL among the active coumarins [29,54].

Evidence from the literature suggests that coumarin itself has a low antibacterial activity, but compounds having long chain hydrocarbon substitutions such as ammoresinol and ostruthin show activity against a wide spectrum of Gram-positive bacteria such as *Bacillus megaterium*, *Micrococcus luteus*, *Micrococcus lysodeikticus*, and *Staphylococcus aureus* [55]. Hydroxycoumarins showed an inhibitory effect on the growth of the Gram-negative bacterium *Ralstonia solanacearum* in a 2016 study [56].

The biological activity of angular pyranocoumarins is poorly investigated [29]. Our investigation of antibacterial activity based on a photometric test indicated that each extract showed more or less pronounced antibacterial potencies, affecting both Gram-positive and Gram-negative microorganisms. Specific bacteriostatic effects of *P. sibiricus* and *P. villosus* root extracts against *E. coli*, *S. enterica*, and *S. aureus* were revealed. In the available literature, there is practically no information about the molecular mechanisms of the antibacterial action of angular pyranocoumarins, except for one work showing the inhibitory effect of chemically synthesized angular coumarins on the growth of different strains of bacteria and fungi [57]. In a study of the linear pyranocoumarins decursin and decursinol angelate from *Angelica gigas*, the authors suggest that the six-membered ring and senecioylic acid-type side chain are closely related to the enhanced antibacterial activities of coumarins against *B. subtilis* [29]. However, it was proposed that pyranocoumarins proved to be significantly less effective than all the other series of coumarin derivatives against all tested microorganisms (*Bacillus cereus* MIP 96016, *Escherichia coli* ATCC 25922, *Pseudomonas aeruginosa* ATCC 27853, and *Staphylococcus aureus* ATCC 25923), suggesting that the pyrano-ring is not required for enhancing the antibacterial activity of the coumarin per se [30].

Pharmacological and clinical studies of furanocoumarins are fragmentary [12]. Historically, furanocoumarins have been used to treat skin diseases. The linear furanocoumarin bergapten and bergamottin absorb light in the near-UV region and stimulate the formation of melanin [58]. This same property of photon absorption leads to the formation of a triplet excited state, which reacts with pyrimidine bases or with oxygen, forming reactive oxygen species. The DNA binding potential, which disrupts the basic biological functions of this macromolecule, is one of the main reasons for the toxicity of furanocoumarins to a wide range of organisms, including mammals, viruses, bacteria, plants, insects, and fungi [12,58,59,60,61,62]. Therefore, furanocoumarins have high toxicity, and as a result, they are widely used medically. Linear furanocoumarin psoralene and its derivatives had antimycobacterial activity and in vitro synergistic activity in combination with antituberculosis drugs (isoniazid, rifampicin, and ethambutol) [12,29]. Moreover, the antimicrobial activity of individual simple coumarin (4′-senecioiloxyosthol) and furanocoumarin (auraptenol) were reported previously against *Bacillus subtilis* (MIC at 5 µg/mL) and *Staphylococcus aureus* (MIC at 16 µg/mL), respectively [54].

Thus, species of the genus *Phlojodicarpus* are not only of practical interest as sources of furano- and pyranocoumarins but also a good model for further study of the biosynthesis of the angular forms of coumarins. The common pathway for the biosynthesis of angular forms in two related species may help identify key enzymes not only in their formation but also enzymes involved in subsequent stages of furano- and pyranocoumarin synthesis. The investigation of the antibacterial activity of the extracts and individual forms of pyrano- and furanocoumarins may be useful for creating coumarin-base derivatives with different structure forms with potential antibacterial activities against a variety of Gram-negative and Gram-positive bacteria. At the same time, for a high yield of substances, plant model objects or their cell cultures are required. *P. sibiricus* and *P. villosus* can serve as models for subsequent experimental work in investigating the biosynthesis of furano- and pyranocoumarins.

## 4. Materials and Methods

### 4.1. Object of Study

For this study, samples were collected in the territory of the Republic of Sakha (Yakutia) of the Russian Federation. Samples of *P. sibiricus* were collected in the floodplain of the Amga River (central Yakutia) (population designation—I) and the indigenous bank of the Chara River (southern Yakutia) (population designation—II). *P. villosus* was collected on the Verkhoyansk ridges in the floodplain of the Tumara River (central Yakutia). All samples were collected in 2021.

### 4.2. Analytical Chromatography and Mass Spectrometry

The air-dried and powdered samples (about 100 mg) were sonicated at 40 °C for 30 min in 2 mL of 80% *v*/*v* methanol, equilibrated for 20 h in darkness at room temperature, and then centrifuged (15,000× *g*, 10 min). The supernatant was filtered and the residue was re-extracted in the same manner. The extracts were combined, cleared with a 0.45 μm membrane (Millipore, Bedford, MA, USA), and used for HPLC analysis.

Reversed-phase high-performance liquid chromatography with the diode array detection and electrospray ionization high-resolution mass spectrometry (RP-HPLC-DAD-ESI-HR-MS/MS^2^) method was applied for the coumarin determination. An LCMS-IT-TOF instrument was used, including the LC-20AD Prominence and ion-trap/time-of-flight mass spectrometer (Shimadzu, Kyoto, Japan). The separation was carried out using an analytical Zorbax C18 column (150 mm, 2.1 mm i.d., 3.5 μm part size, Agilent Technologies, Santa Clara, CA, USA) with the flow rate of 0.2 mL/min and the column temperature at 40 °C. A binary gradient with A (of 0.1% aqueous formic acid) and B (acetonitrile) was installed starting with 0% B. The following gradient profile was used: 0–10 min linear gradient from 0% to 15% B; 10–25 min isocratic on 15% B; 25–40 min linear gradient from 15% to 50% B; 40–55 min isocratic on 50% B; 55–65 min linear gradient from 50% to 60% B; 65–80 min isocratic on 60% B; and 80–100 min linear gradient from 60% to 100% B. UV/Vis spectra were recorded with a DAD in the range of 200–800 nm and chromatograms were obtained at a wavelength of 320 nm. Mass spectra were recorded with the simultaneous registration of negative and positive ions and a resolution of 12,000. The following settings were used: the range of *m/z* detection was 100–1200, the drying gas (N_2_) pressure was 150 kPa, the nebulizer gas flow rate was 1.5 L/min, the ion source potential changed from −3.8 to 4.5 kV, and the interface temperature was 200 °C. Also, MS^2^ data were collected in automatic mode, the precursor isolation width was set at 0.05 *m/z*, and collision energy and collision gas were set at 50%. All MS data were collected and processed using the Shimadzu LCMS Solution software (v.3.60.361). Coumarin identification was carried out using chromatographic behavior, UV, and mass-spectral data of the studied compounds and comparison with reference standards and data from the literature. The quantification of all identified compounds was performed by peak areas determined at 320 nm using the external standard calibration method with authentic standards of visnadine (Sigma, St. Louis, MI, USA) and peucenidin (previously isolated from *Ferulopsis hystrix* [34]).

### 4.3. Photometric Bacteriostatic Test

Air-dried roots of *P. sibiricus* (population II) and *P. villosus* were homogenized to a powdery state. A total of 300 mg of the biomass was transferred to a 15 mL Eppendorf tube and poured into 6 mL of 70% ethyl alcohol. The extraction was carried out in an ultrasonic bath for 30 min. The extract was precipitated on an Eppendorf 5910 R centrifuge (Eppendorf, Hamburg, Germany) for 10 min at 15,000× *g*. Each 500 µL of supernatant was transferred to a clean 1.5 mL Eppendorf tube (a total of 12 tubes were obtained). The resulting extract was evaporated in an Eppendorf Vacufuge Plus Concentrator Complete System (Eppendorf, Germany). The dry extract was dissolved in DMSO with distilled sterile water (500 μL). The concentration of DMSO in the test extract was 0.01%. A solution of 25 mg of dry biomass in 1 mL was used for the test.

Conditionally pathogenic Gram-negative bacteria (*E. coli* ATCC 25922, *P. aeruginosa* ATCC 9027, *S. enterica* serovar Abony 103/39) and Gram-positive bacteria (*S. aureus* ATCC 6538P=FDA 209P, *S. epidermidis* ATCC 14990) were used as test microorganisms. For the experiment, the microorganisms were cultured overnight in liquid LB (Lisogeny broth) medium (peptone 10 g/L, yeast extract 5 g/L, NaCl 5 g/L, 1M NaOH 2 mL, pH = 7.0) under continuous circular agitation at 37 °C.

The optical density of the bacterial suspensions was measured using a UV/Vis spectrophotometer NanoPhotometer Pearl (Implen, Munich, Germany) at 600 nm wavelength. They were then diluted in PBM medium (peptone 15 g/L, NaCl 9 g/L) to A600 = 0.05 and used in the experiment. The photometric test of the bacteriostatic effect of the extracts was performed using The BioTek Cytation 5 plate reader with Gen 5 software (BioTek Instruments, Charlotte, VT, USA). Of a flat-bottomed 96-well plate, 8 wells were used for one strain of bacteria for each separate type of experimental sample and control. Measurements were performed in Endpoint mode at 600 nm wavelength overnight at 30 min intervals at a 37 °C incubation temperature with mechanical agitation before each measurement. The optical density of bacterial growth for plotting was counted as follows: for experimental data, the optical density of the extract was subtracted separately for each measurement point; for negative control, the optical density of the solvent was subtracted; and for positive control, the optical density of the antibiotic was subtracted. Bacteria with 0.01% DMSO solution added were used as negative control. A solution of cefotaxime antibiotic at a concentration of 25 mg/mL was used as a positive control.

### 4.4. Statistical Analysis

The Shapiro–Wilk test was used to check the data for normal distribution, and non-parametric methods were chosen as a result. To evaluate the differences between two independent samplings, the Mann–Whitney U test was used. The number of repetitions for each sample was 16 in DDT and 49 in PBT. The analysis was conducted using the software package for statistical analysis GraphPad Prism 6 (GraphPad Holdings, Boston, MA, USA).

## 5. Conclusions

The wide variety of coumarins’ biological activities has expanded the possibilities of using these substances in medicine. The search for new sources of rare groups of useful substances is always relevant. The present study reports great differences in the composition of the coumarin groups of two closely related species, *P. sibiricus* and *P. villosus*. The predominance of pyranocoumarins in *P. sibiricus* was confirmed, and a difference in the qualitative composition of the pyranocoumarins depending on the region of plant growth was found. Furano- and hydroxycoumarins were revealed in *P. villosus*. Using the photometric method, the bacteriostatic effect of *P. sibiricus* and *P. villosus* root extracts against *S. enterica* and *S. aureus* was detected, as well as the inhibition of the growth of *E. coli* bacteria by the root extract of *P. villosus*. Thus, *P. sibiricus* and *P. villosus* are promising sources of biologically active substances and potential models for investigating the biosynthesis pathways of various coumarins, and would provide a framework for pharmacophore-based drug discovery against bacterial diseases.

## Figures and Tables

**Figure 1 plants-13-00601-f001:**
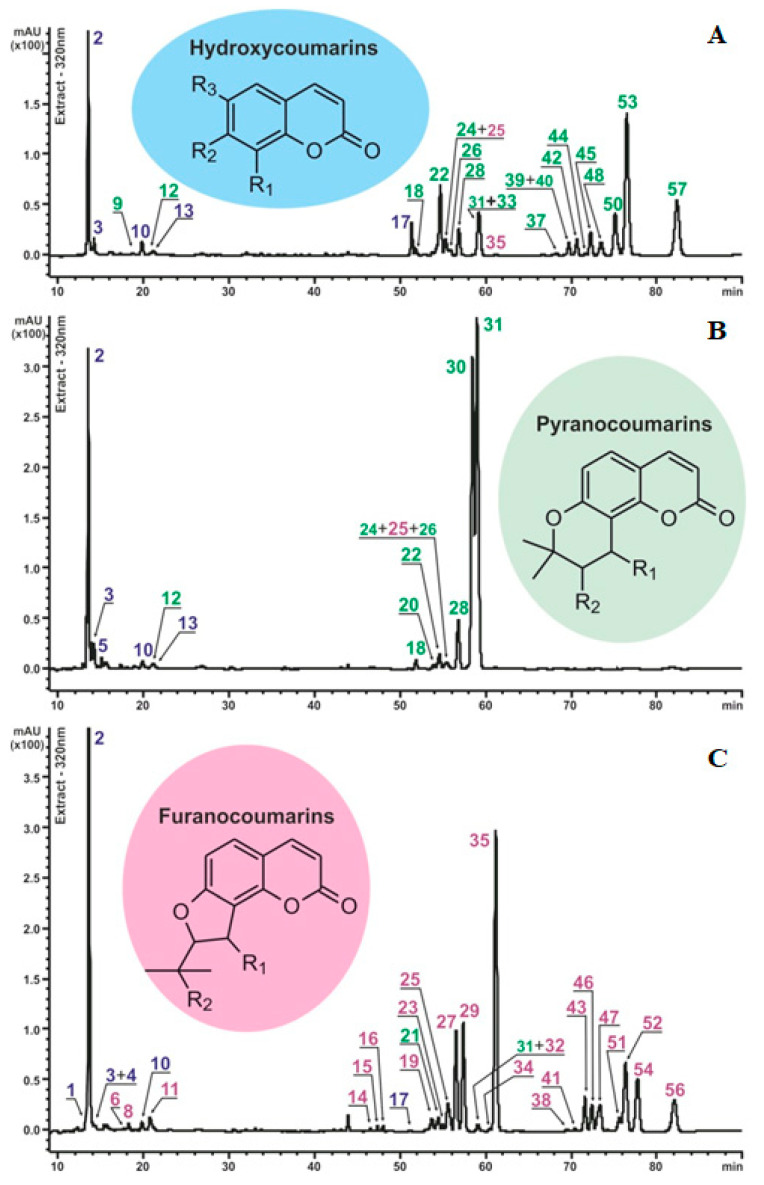
Chromatographic profiles of the methanolic extracts of *P. sibiricus* and *P. villosus* with schematic aglycone structures of the identified compounds. The peak numbers correspond to the components listed in Table 1 and are marked in different colors: blue—hydroxycoumarins (HC); green—pyranocoumarins (PC); pink—furanocoumarins (FC). Chromatograms: (**A**)—*P. sibiricus* population I; (**B**)—*P. sibiricus* population II; (**C**)—*P. villosus*. The measurement was carried out at a wavelength of 320 nm.

**Figure 2 plants-13-00601-f002:**
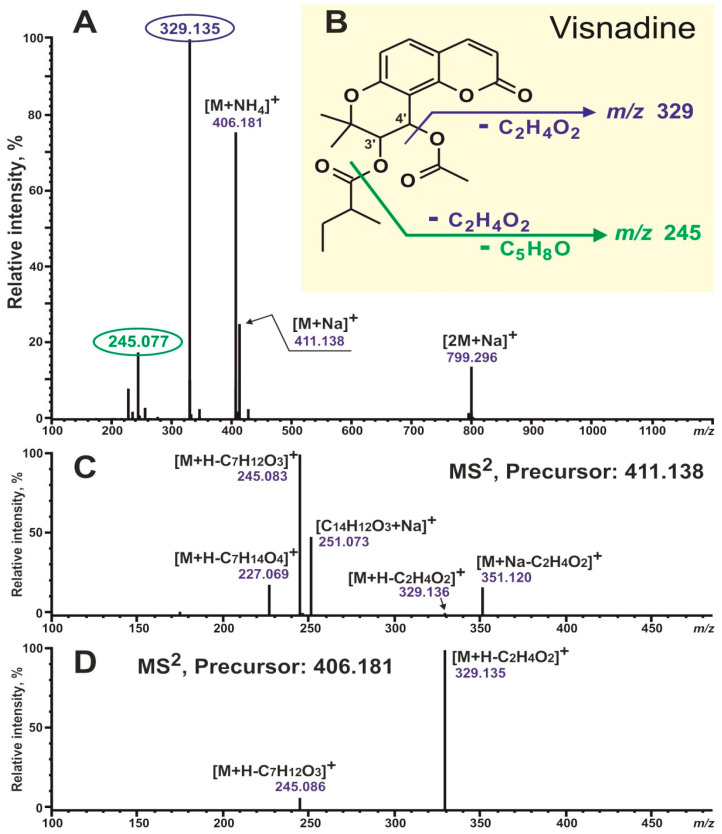
Mass spectra and fragmentation pattern of reference visnadine, recorded using HPLC-ESI-HR-MS/MS^2^. (**A**) Positive full scan spectrum. (**B**) The possible fragmentation pathway scheme of visnadine. (**C**) MS^2^ fragmentation spectrum of precursor ions [M+Na]^+^. (**D**) MS^2^ fragmentation spectrum of precursor ions [M+NH_4_]^+^.

**Figure 3 plants-13-00601-f003:**
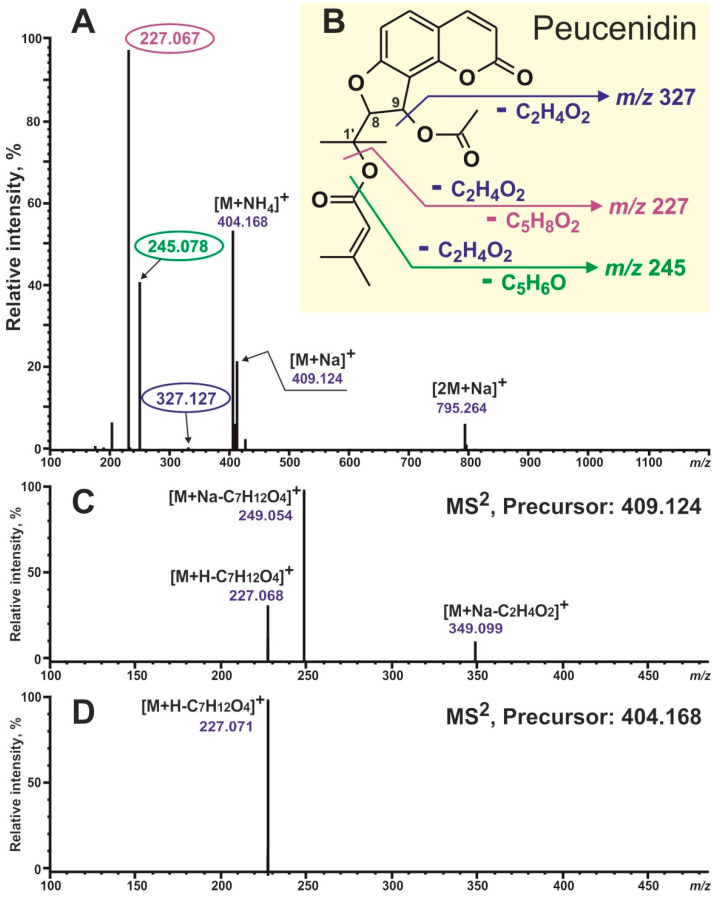
Mass spectra and fragmentation pattern of authentic peucenidin, recorded using HPLC-ESI-HR-MS/MS^2^. (**A**) Positive full scan spectrum. (**B**) The possible fragmentation pathway scheme of peucenidin. (**C**) MS^2^ fragmentation spectrum of precursor ions [M+Na]^+^. (**D**) MS^2^ fragmentation spectrum of precursor ions [M+NH_4_]^+^.

**Figure 4 plants-13-00601-f004:**
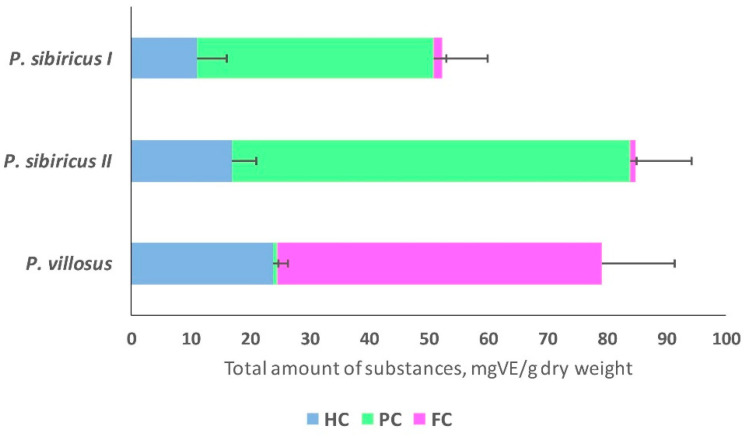
The content of different groups of coumarins in *P. sibiricus* (populations I and II) and *P. villosus* extracts. Blue HC—hydroxycoumarins; green PC—pyranocoumarins; pink FC—furanocoumarins. The total coumarin content of the different groups was calculated as mgVE/g dry weight for pyranocoumarins and mgPE/g dry weight for furanocoumarins (VE—visnadine equivalent; PE—peucenidin equivalent), further in the text mg/g.

**Figure 5 plants-13-00601-f005:**
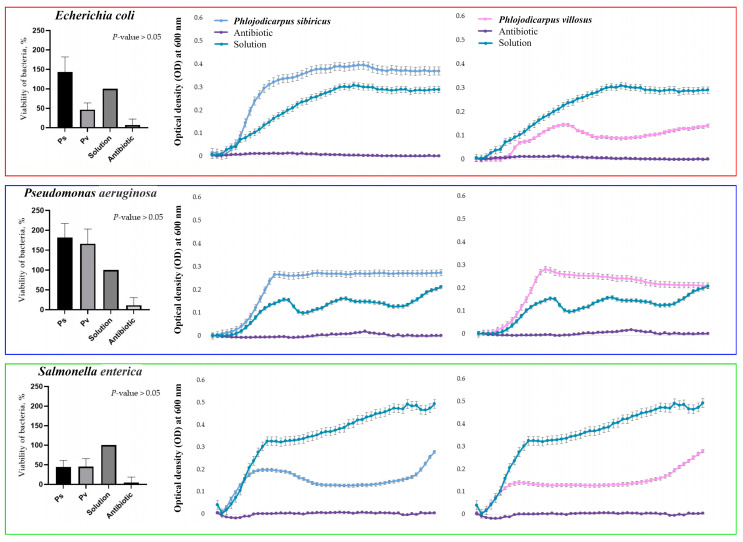
Bacteriostatic effect of 0.01% DMSO extracts of *P. sibiricus* and *P. villosus* against Gram-negative bacteria. The points on the horizontal axis correspond to measurements every 30 min for 24 h. To construct viability histograms, the values of the negative control were compared with the values of the samples and the effect of the antibiotic. Therefore, bacterial growth in the medium containing DMSO was taken as 100%, so there is no standard deviation for this column.

**Figure 6 plants-13-00601-f006:**
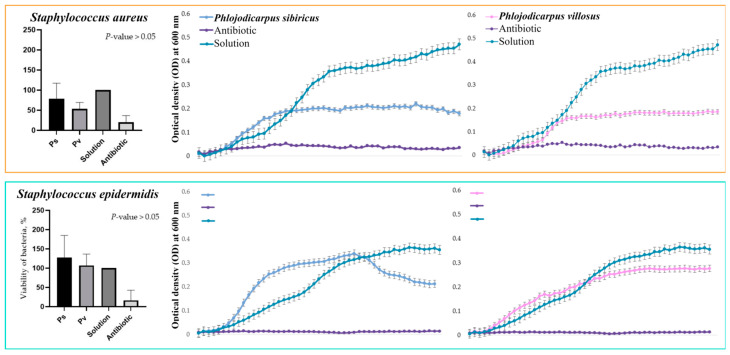
Bacteriostatic effect of 0.01% DMSO extracts of *P. sibiricus* and *P. villosus* against Gram-positive bacteria. The points on the horizontal axis correspond to measurements every 30 min for 24 h. To construct viability histograms, the values of the negative control were compared with the values of the samples and the effect of the antibiotic. Therefore, bacterial growth in the medium containing DMSO was taken as 100%, so there is no standard deviation for this column.

**Figure 7 plants-13-00601-f007:**
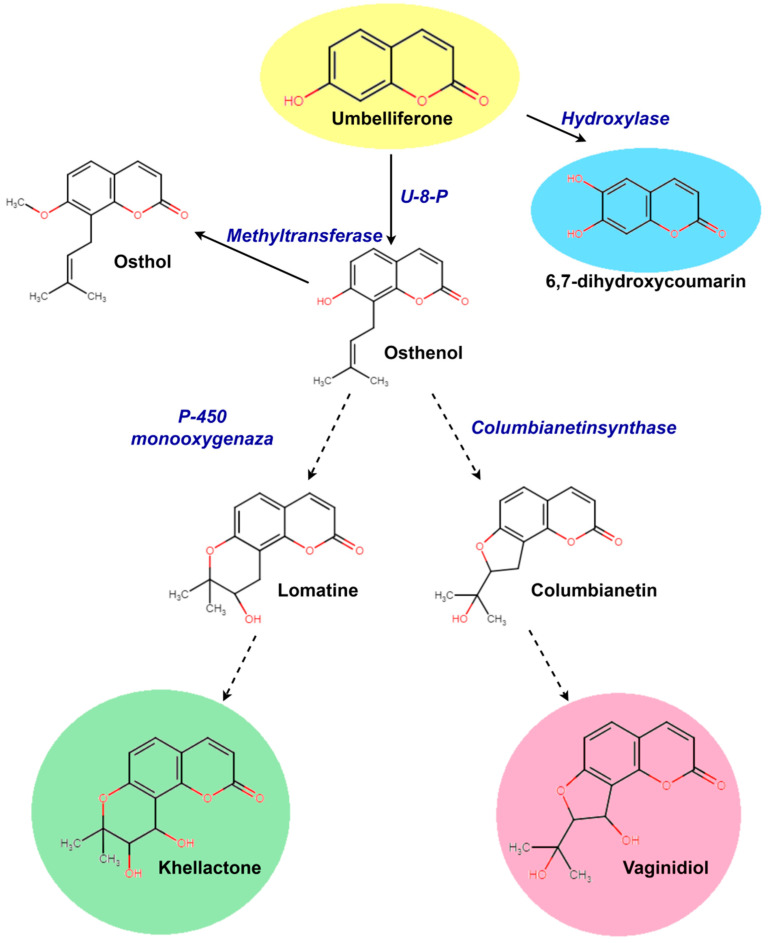
The scheme for the biosynthesis of dihydro-, angular pyrano-, and furanocoumarins from umbelliferone [41,42,43,44,45,46].

**Table 1 plants-13-00601-t001:** List of coumarin compounds identified in root extracts *P. sibiricus* and *P. villosus* using HPLC-UV-ESI-MS/MS^2^.

No	T_R_, min	Compound ^1^	UV max, nm	Molecular Formula (m.v.)	ESI-MS(*MS^2^*) Data, Main Diagnostic Ions	
					Ion Composition *(for MS^2^: all MS^2^ information is displayed in italics, Precursor Ion Composition is Shown in Parentheses)*	*m*/*z* Values
1	13.2	Umbelliferone-*O*-pentosyl-*O*-hexoside ^2^	317	C_20_H_24_O_12_ (456)	[M+Na]^+^	479
					[M+H-C_5_H_8_O_4_]^+^	325
					[M-H+Fa]^−^	501
					[M-H]^−^	455
2	13.6	Umbelliferone-*O*-pentosyl-*O*-hexoside (Apiosylskimmin) ^2^	317	C_20_H_24_O_12_ (456)	[M+Na]^+^	479
					[M+H-C_5_H_8_O_4_]^+^	325
					[M+H-C_5_H_8_O_4_-C_6_H_10_O_5_]^+^	163
					[M-H+Fa]^−^	501
					[M-H]^−^	455
					*MS^2^(* *[M-H]* ^−^ *): [M-H-C_9_H_6_O_3_]* * ^−^ *	*293*
3	14.2	Hydroxymethoxycoumarin-*O*-pentosyl-*O*-hexoside (Scopoletin-*O*-pentosyl-*O*-hexoside) ^2^	290, 332	C_21_H_26_O_13_ (486)	[M+Na]^+^	509
					[M+H-C_5_H_8_O_4_]^+^	355
					[M-H+Fa]^−^	531
					[M-H]^−^	485
					*MS^2^(* *[M-H]^−^): [M-H-C_10_H_8_O_4_]^−^*	*293*
4	14.4	Hydroxydimethoxycoumarin-*O*-pentosyl-*O*-hexoside (Fraxidin-*O*-pentosyl-*O*-hexoside) ^2^	288, 326	C_22_H_28_O_14_ (516)	[M+Na]^+^	539
					[M-H+Fa]^−^	561
					[M-H]^−^	515
					*MS^2^(* *[M-H]^−^): [M-H-C_11_H_18_O_9_]^−^*	*221*
5	15.1	Peucedanol-*O*-pentosyl-*O*-hexoside ^2^	296, 325	C_25_H_34_O_14_ (558)	[M+Na]^+^	581
					[M+H-C_5_H_8_O_4_]^+^	427
					[M+H-C_5_H_8_O_4_-C_6_H_10_O_5_]^+^	265
					[M-H+Fa]^−^	603
					[M-H]^−^	557
					*MS^2^(* *[M-H]^−^): [M-H-C_11_H_18_O_9_]^−^*	*263*
6	17.7	Vaginidiol-*O*-pentosyl-*O*-hexoside ^2^	329	C_25_H_32_O_14_ (556)	[M+Na]^+^	579
					[M+H-C_5_H_8_O_4_-C_6_H_10_O_5_]^+^	263
					[M-H+Fa]^−^	601
					[M-H]^−^	555
7	17.9	Khellactone-*O*-pentosyl-*O*-hexoside ^2^	nd	C_25_H_32_O_14_ (556)	[M+Na]^+^	579
					[M+H-C_5_H_8_O_4_-C_6_H_10_O_5_]^+^	263
					[M-H+Fa]^−^	601
					[M-H]^−^	555
8	18.2	Vaginidiol-*O*-hexoside ^2^	332	C_20_H_24_O_10_ (424)	[M+Na]^+^	447
					[M+H-C_5_H_8_O_4_-C_6_H_10_O_5_]^+^	263
					[M-H+Fa]^−^	469
					[M-H]^−^	423
					*MS^2^(* *[M+H-C_5_H_8_O_4_-C_6_H_10_O_5_]^+^): H_2_O loss*	*245*
					*MS^2^(* *[M+H-C_5_H_8_O_4_-C_6_H_10_O_5_]^+^):2H_2_O loss*	*227*
9	18.4	Khellactone-*O*-hexoside ^2^	332	C_20_H_24_O_10_ (424)	[M+Na]^+^	447
					[M+H-C_5_H_8_O_4_-C_6_H_10_O_5_]^+^	263
					[M-H+Fa]^−^	469
					*MS^2^(* *[M+H-C_5_H_8_O_4_-C_6_H_10_O_5_]^+^): H_2_O loss*	*245*
10	19.8	Umbelliferone ^2^	324	C_9_H_6_O_3_ (162)	[M+H]^+^	163
11	20.7	Vaginidiol-*O*-hexoside ^2^	323	C_20_H_24_O_10_ (424)	[2M+Na]^+^	871
					[M+Na]^+^	447
					[M-H+Fa]^−^	469
					[M-H]^−^	423
12	21.0	Khellactone-*O*-hexoside (Praeroside II) ^2^	324	C_20_H_24_O_10_ (424)	[2M+Na]^+^	871
					[M+Na]^+^	447
					[M-H+Fa]^−^	469
					[M-H]^−^	423
13	21.3	Peucedanol-*O*-glucoside ^2^	326	C_20_H_26_O_10_ (426)	[M+H]^+^	427
					[M+H-C_6_H_10_O_5_]^+^	265
					[M-H]^−^	425
14	46.4	Vaginidiol monoacyl ester ^2^ R1 or R2–C_5_H_7_O_2_ ^3^ R2 or R1–OH	323	C_19_H_20_O_6_ (344)	[2M+Na]^+^	711
					[M+Na]^+^	367
					[M+NH_4_]^+^	362
					[M+H-C_5_H_8_O_2_]^+^	245
					[M+H-C_5_H_8_O_2_-H_2_O]^+^	227
					*MS^2^(* *[M+Na]^+^): [M+Na-C_5_H_8_O_2_]^+^*	*267*
15	47.3	Vaginidiol monoacyl ester ^2^ R1 or R2–C_5_H_7_O_2_ ^3^ R2 or R1–OH	322	C_19_H_20_O_6_ (344)	[2M+Na]^+^	711
					[M+Na]^+^	367
					[M+NH_4_]^+^	362
					[M+H-C_5_H_8_O_2_]^+^	245
					[M+H-C_5_H_8_O_2_-H_2_O]^+^	227
					*MS^2^(* *[M+Na]^+^): [M+Na-C_5_H_8_O_2_]^+^*	*267*
16	47.9	Vaginidiol monoacyl ester ^2^ R1 or R2–C_5_H_9_O_2_ ^3^ R2 or R1–OH	326	C_19_H_22_O_6_ (346)	[2M+Na]^+^	715
					[M+Na]^+^	369
					[M+NH_4_]^+^	364
					[M+H-C_5_H_10_O_2_]^+^	245
					[M+H-C_5_H_10_O_2_-H_2_O]^+^	227
					*MS^2^(* *[M+Na]^+^): [M+Na-C_5_H_10_O_2_]^+^*	*267*
17	51.3	Methoxycoumarin prenylated (Osthole) ^2^	322	C_15_H_16_O_3_ (244)	[M+H]^+^	245
					[M+H-C_4_H_8_]^+^	189
18	51.8	Khellactone diacyl ester ^2^ R1–C_4_H_7_O_2_ ^3^ R2–C_2_H_3_O_2_ ^3^ (Hyuganin D) ^2^	321	C_20_H_22_O_7_ (374)	[2M+Na]^+^	771
					[M+Na]^+^	397
					[M+NH_4_]^+^	392
					[M+H-C_4_H_8_O_2_]^+^	287
					[M+H-C_4_H_8_O_2_-C_2_H_2_O]^+^	245
19	53.7	Vaginidiol monoacyl ester ^2^ R1 or R2–C_5_H_7_O_2_ ^3^ R2 or R1–H (Libanorin/Columbianadin) ^2^	328	C_19_H_20_O_5_ (328)	[M-H]^+^	329
					[M+H-C_5_H_8_O_2_]^+^	229
20	53.9	Khellactone diacyl ester ^2^ R1–C_2_H_3_O_2_ ^3^ R2–C_5_H_7_O_2_ ^3^	323	C_21_H_22_O_7_ (386)	[2M+Na]^+^	795
					[M+Na]^+^	409
					[M+NH_4_]^+^	404
					[M+H-C_2_H_4_O_2_]^+^	327
21	54.4	Khellactone diacyl ester ^2^ R1–C_2_H_3_O_2_ ^3^ R2–C_5_H_7_O_2_ ^3^	323	C_21_H_22_O_7_ (386)	[2M+Na]^+^	795
					[M+Na]^+^	409
					[M+NH_4_]^+^	404
					[M+H-C_2_H_4_O_2_]^+^	327
22	54.6	Khellactone diacyl ester ^2^ R1–C_5_H_7_O_2_ ^3^ R2–C_2_H_3_O_2_ ^3^ (Pteryxin ^2^)	322	C_21_H_22_O_7_ (386)	[2M+Na]^+^	795
					[M+Na]^+^	409
					[M+NH_4_]^+^	404
					[M+H-C_5_H_8_O_2_]^+^	287
					[M+H-C_5_H_8_O_2_-C_2_H_2_O]^+^	245
23	54.9	Vaginidiol diacyl ester ^2^ R1–C_5_H_7_O_2_ ^3^ R2–C_2_H_3_O_2_ ^3^	323	C_21_H_22_O_7_ (386)	[2M+Na]^+^	795
					[M+Na]^+^	409
					[M+NH_4_]^+^	404
					[M+H-C_5_H_8_O_2_-C_2_H_2_O]^+^	245
					[M+H-C_5_H_8_O_2_-C_2_H_4_O_2_]^+^	227
					*MS^2^(* *[M+Na]^+^): [M+Na-C_5_H_8_O_2_]^+^*	*309*
24	55.3	Khellactone monoacyl ester ^2^ R1 or R2–C_5_H_7_O_2_ ^3^ R2 or R1–H (Lomatin-*O*-senecioyl ester) ^2^	325	C_19_H_20_O_5_ 328	[M+Na]^+^	351
					[M+H]^+^	329
					[M+H-C_5_H_8_O_2_]^+^	229
25	55.5	Vaginidiol diacyl ester ^2^ R1–C_5_H_7_O_2_ ^3^ R2–C_2_H_3_O_2_ ^3^	322	C_21_H_22_O_7_ (386)	[2M+Na]^+^	795
					[M+Na]^+^	409
					[M+NH_4_]^+^	404
					[M+H-C_5_H_8_O_2_-C_2_H_2_O]^+^	245
					[M+H-C_5_H_8_O_2_-C_2_H_4_O_2_]^+^	227
					*MS^2^(* *[M+Na]^+^): [M+Na-C_5_H_8_O_2_]^+^*	*309*
26	55.9	Khellactone diacyl ester ^2^ R1–C_2_H_3_O_2_ ^3^ R2–C_5_H_7_O_2_ ^3^	321	C_21_H_22_O_7_ (386)	[2M+Na]^+^	795
					[M+Na]^+^	409
					[M+NH_4_]^+^	404
					[M+H-C_2_H_4_O_2_]^+^	327
27	56.5	Peucenidin ^4^	322	C_21_H_22_O_7_ (386)	[2M+Na]^+^	795
					[M+Na]^+^	409
					[M+NH_4_]^+^	404
					[M+H-C_5_H_6_O-C_2_H_4_O_2_]^+^	245
					[M+H-C_5_H_8_O_2_-C_2_H_4_O_2_]^+^	227
					*MS^2^(* *[M+Na]^+^): [M+Na-C_2_H_4_O_2_]^+^*	*349*
28	56.8	Khellactone monoacyl ester ^2^ R1 or R2–C_5_H_9_O_2_ ^3^ R2 or R1–H (Lomatin-*O*-isovaleroyl ester) ^2^	325	C_19_H_22_O_5_ (330)	[M+Na]^+^	353
					[M+H]^+^	331
					[M+H-C_5_H_10_O_2_]^+^	229
29	57.3	Vaginidiol diacyl ester ^2^ R1–C_5_H_7_O_2_ ^3^ R2–C_2_H_3_O_2_ ^3^ (Libanotin) ^2^	322	C_21_H_22_O_7_ (386)	[2M+Na]^+^	795
					[M+Na]^+^	409
					[M+NH_4_]^+^	404
					[M+H-C_5_H_8_O_2_-C_2_H_4_O]^+^	245
					[M+H-C_5_H_8_O_2_-C_2_H_4_O_2_]^+^	227
					*MS^2^(* *[M+Na]^+^): [M+Na-C_5_H_8_O_2_]^+^*	*309*
30	58.5	Visnadin ^4^	323	C_21_H_24_O_7_ (388)	[2M+Na]^+^	799
					[M+Na]^+^	411
					[M+NH_4_]^+^	406
					[M+H-C_2_H_4_O_2_]^+^	329
					[M+H-C_2_H_4_O_2_-C_5_H_8_O]^+^	245
31	59.0	Khellactone diacyl ester ^2^ R1–C_2_H_3_O_2_ ^3^ R2–C_5_H_9_O_2_ ^3^ (Dihydrosamidin) ^2^	322	C_21_H_24_O_7_ (388)	[2M+Na]^+^	799
					[M+Na]^+^	411
					[M+NH_4_]^+^	406
					[M+H-C_2_H_4_O_2_]^+^	329
					[M+H-C_2_H_4_O_2_-C_5_H_8_O]^+^	245
32	59.1	Vaginidiol diacyl ester ^2^ R1–C_2_H_3_O_2_ ^3^ R2–C_5_H_9_O_2_ ^3^	319	C_21_H_24_O_7_ (388)	[2M+Na]^+^	799
					[M+Na]^+^	411
					[M+NH_4_]^+^	406
					[M+H-C_2_H_4_O_2_-C_5_H_8_O]^+^	245
					[M+H-C_5_H_10_O_2_-C_2_H_4_O_2_]^+^	227
					*MS^2^(* *[M+Na]^+^): [M+Na-C_2_H_4_O_2_]^+^*	*351*
33	59.2	Khellactone diacyl ester ^2^ R1–C_5_H_9_O_2_ ^3^ R2–C_2_H_3_O_2_ ^3 ^ (Suksdorfin) ^2^	322	C_21_H_24_O_7_ (388)	[2M+Na]^+^	799
					[M+Na]^+^	411
					[M+NH_4_]^+^	406
					[M+H-C_5_H_10_O_2_]^+^	287
					[M+H-C_2_H_2_O-C_5_H_10_O_2_]^+^	245
34	60.4	Vaginidiol diacyl ester ^2^ R1–C_2_H_3_O_2_ ^3^ R2–C_5_H_9_O_2_ ^3^	323	C_21_H_24_O_7_ (388)	[2M+Na]^+^	799
					[M+Na]^+^	411
					[M+NH_4_]^+^	406
					[M+H-C_2_H_4_O_2_-C_5_H_8_O]^+^	245
					[M+H-C_5_H_10_O_2_-C_2_H_4_O_2_]^+^	227
					*MS^2^(* *[M+Na]^+^): [M+Na-C_2_H_4_O_2_]^+^*	*351*
35	61.1	Vaginidiol diacyl ester ^2^ R1–C_2_H_3_O_2_ ^3^ R2–C_5_H_9_O_2_ ^3^	321	C_21_H_24_O_7_ (388)	[2M+Na]^+^	799
					[M+Na]^+^	411
					[M+NH_4_]^+^	406
					[M+H-C_2_H_4_O_2_-C_5_H_8_O]^+^	245
					[M+H-C_5_H_10_O_2_-C_2_H_4_O_2_]^+^	227
					*MS^2^(* *[M+Na]^+^): [M+Na-C_2_H_4_O_2_]^+^*	*351*
36	61.1	Khellactone diacyl ester ^2^ R1–C_2_H_3_O_2_ ^3^ R2–C_5_H_9_O_2_ ^3^	323	C_21_H_24_O_7_ (388)	[2M+Na]^+^	799
					[M+Na]^+^	411
					[M+NH_4_]^+^	406
					[M+H-C_2_H_4_O_2_]^+^	329
					[M+H-C_2_H_4_O_2_-C_5_H_8_O]^+^	245
37	68.3	Khellactone diacyl ester ^2^ R1–C_4_H_7_O_2_ ^3^ R2–C_5_H_7_O_2_ ^3^	320	C_23_H_26_O_7_ (414)	[2M+Na]^+^	851
					[M+Na]^+^	437
					[M+NH_4_]^+^	432
					[M+H-C_4_H_8_O_2_]^+^	327
38	69.4	Vaginidiol diacyl ester ^2^ R1–C_4_H_7_O_2_ ^3^ R2–C_5_H_7_O_2_ ^3^ together with R1–C_5_H_7_O_2_ ^3^ R2–C_4_H_7_O_2_ ^3^	321	C_23_H_26_O_7_ (414)	[2M+Na]^+^	851
					[M+Na]^+^	437
					[M+NH_4_]^+^	432
					[M+H-C_4_H_8_O_2_-C_5_H_6_O]^+^	245
					[M+H-C_4_H_8_O_2_-C_5_H_8_O_2_]^+^	227
					*MS^2^(* *[M+Na]^+^): [M+Na-C_4_H_8_O_2_]^+^*	*349*
					*MS^2^(* *[M+Na]^+^): [M+Na-C_5_H_8_O_2_]^+^*	*337*
39	69.7	Khellactone diacyl ester ^2^ R1–C_4_H_7_O_2_ ^3^ R2–C_5_H_7_O_2_ ^3^	320	C_23_H_26_O_7_ (414)	[2M+Na]^+^	851
					[M+Na]^+^	437
					[M+NH_4_]^+^	432
					[M+H-C_4_H_8_O_2_]^+^	327
40	69.7	Khellactone diacyl ester ^2^ R1–C_5_H_7_O_2_ ^3^ R2–C_5_H_7_O_2_ ^3^	323	C_24_H_26_O_7_ (426)	[2M+Na]^+^	875
					[M+Na]^+^	449
					[M+NH_4_]^+^	444
					[M+H-C_5_H_8_O_2_]^+^	327
41	70.4	Vaginidiol diacyl ester ^2^ R1–C_5_H_7_O_2_ ^3^ R2–C_4_H_7_O_2_ ^3^	320	C_23_H_26_O_7_ (414)	[2M+Na]^+^	851
					[M+Na]^+^	437
					[M+H-C_5_H_8_O_2_-C_4_H_6_O]^+^	245
					[M+H-C_4_H_8_O_2_-C_5_H_8_O_2_]^+^	227
					*MS^2^(* *[M+Na]^+^): [M+Na-C_5_H_8_O_2_]^+^*	*337*
42	70.6	Khellactone diacyl ester ^2^ R1–C_5_H_7_O_2_ ^3^ R2–C_5_H_7_O_2_ ^3^	322	C_24_H_26_O_7_ (426)	[2M+Na]^+^	875
					[M+Na]^+^	449
					[M+NH_4_]^+^	444
					[M+H-C_5_H_8_O_2_]^+^	327
43	71.5	Vaginidiol diacyl ester ^2^ R1–C_5_H_7_O_2_ ^3^ R2–C_5_H_7_O_2_ ^3^	321	C_24_H_26_O_7_ (426)	[2M+Na]^+^	875
					[M+Na]^+^	449
					[M+NH_4_]^+^	444
					[M+H-C_5_H_8_O_2_-C_5_H_6_O]^+^	245
					[M+H-C_5_H_8_O_2_-C_5_H_8_O_2_]^+^	227
					*MS^2^(* *[M+Na]^+^): [M+Na-C_5_H_8_O_2_]^+^*	*349*
44	71.6	Khellactone diacyl ester ^2^ R1–C_5_H_7_O_2_ ^3^ R2–C_5_H_7_O_2_ ^3^	322	C_24_H_26_O_7_ (426)	[2M+Na]^+^	875
					[M+Na]^+^	449
					[M+NH_4_]^+^	444
					[M+H-C_5_H_8_O_2_]^+^	327
45	72.2	Khellactone diacyl ester ^2^ R1–C_5_H_7_O_2_ ^3^ R2–C_5_H_7_O_2_ ^3^ (Pracruptorin D) ^2^	321	C_24_H_26_O_7_ (426)	[2M+Na]^+^	875
					[M+Na]^+^	449
					[M+NH_4_]^+^	444
					[M+H-C_5_H_8_O_2_]^+^	327
46	72.4	Vaginidiol diacyl ester ^2^ R1–C_5_H_7_O_2_ ^3^ R2–C_5_H_7_O_2_ ^3^	322	C_24_H_26_O_7_ (426)	[2M+Na]^+^	875
					[M+Na]^+^	449
					[M+NH_4_]^+^	444
					[M+H-C_5_H_8_O_2_-C_5_H_6_O]^+^	245
					[M+H-C_5_H_8_O_2_-C_5_H_8_O_2_]^+^	227
					*MS^2^(* *[M+Na]^+^): [M+Na-C_5_H_8_O_2_]^+^*	*349*
47	73.3	Vaginidiol diacyl ester ^2^ R1–C_5_H_7_O_2_ ^3^ R2–C_5_H_7_O_2_ ^3^	322	C_24_H_26_O_7_ (426)	[2M+Na]^+^	875
					[M+Na]^+^	449
					[M+NH_4_]^+^	444
					[M+H-C_5_H_8_O_2_-C_5_H_6_O]^+^	245
					[M+H-C_5_H_8_O_2_-C_5_H_8_O_2_]^+^	227
					*MS^2^(* *[M+Na]^+^): [M+Na-C_5_H_8_O_2_]^+^*	*349*
48	73.4	Khellactone diacyl ester ^2^ R1–C_4_H_7_O_2_ ^3^ R2–C_5_H_9_O_2_ ^3^	321	C_23_H_28_O_7_ (416)	[2M+Na]^+^	855
					[M+Na]^+^	439
					[M+NH_4_]^+^	434
					[M+H-C_4_H_8_O_2_]^+^	329
49	74.2	Vaginidiol diacyl ester ^2^ R1–C_4_H_7_O_2_ ^3^ R2–C_5_H_9_O_2_ ^3^	323	C_23_H_28_O_7_ (416)	[2M+Na]^+^	855
					[M+Na]^+^	439
					[M+NH_4_]^+^	434
					[M+H-C_4_H_8_O_2_-C_5_H_8_O]^+^	245
					[M+H-C_4_H_8_O_2_-C_5_H_10_O_2_]^+^	227
					*MS^2^(* *[M+Na]^+^): [M+Na-C_4_H_8_O_2_]^+^*	*351*
50	75.1	Khellactone diacyl ester ^2^ R1–C_5_H_7_O_2_ ^3^ R2–C_5_H_9_O_2_ ^3^	323	C_24_H_28_O_7_ (428)	[2M+Na]^+^	879
					[M+Na]^+^	451
					[M+NH_4_]^+^	446
					[M+H-C_5_H_8_O_2_]^+^	329
51	75.6	Vaginidiol diacyl ester ^2^ R1–C_5_H_9_O_2_ ^3^ R2–C_5_H_7_O_2_ ^3^	321	C_24_H_28_O_7_ (428)	[2M+Na]^+^	879
					[M+Na]^+^	451
					[M+NH_4_]^+^	446
					[M+H-C_5_H_10_O_2_-C_5_H_6_O]^+^	245
					[M+H-C_5_H_10_O_2_-C_5_H_8_O_2_]^+^	227
					*MS^2^(* *[M+Na]^+^): [M+Na-C_5_H_10_O_2_]^+^*	*349*
52	76.3	Vaginidiol diacyl ester ^2^ R1–C_5_H_7_O_2_ ^3^ R2–C_5_H_9_O_2_ ^3^	322	C_24_H_28_O_7_ (428)	[2M+Na]^+^	879
					[M+Na]^+^	451
					[M+NH_4_]^+^	446
					[M+H-C_5_H_8_O_2_-C_5_H_8_O]^+^	245
					[M+H-C_5_H_8_O_2_-C_5_H_10_O_2_]^+^	227
					*MS^2^(* *[M+Na]^+^): [M+Na-C_5_H_8_O_2_]^+^*	*351*
53	76.5	Khellactone diacyl ester ^2^ R1–C_5_H_7_O_2_ ^3^ R2–C_5_H_9_O_2_ ^3^	323	C_24_H_28_O_7_ (428)	[2M+Na]^+^	879
					[M+Na]^+^	451
					[M+NH_4_]^+^	446
					[M+H-C_5_H_8_O_2_]^+^	329
					[M+H-C_5_H_8_O_2_-C_5_H_8_O]^+^	245
54	77.7	Vaginidiol diacyl ester ^2^ R1–C_5_H_7_O_2_^3^ R2–C_5_H_9_O_2_^3^	320	C_24_H_28_O_7_ (428)	[2M+Na]^+^	879
					[M+Na]^+^	451
					[M+NH_4_]^+^	446
					[M+H-C_5_H_8_O_2_-C_5_H_8_O]^+^	245
					[M+H-C_5_H_8_O_2_-C_5_H_10_O_2_]^+^	227
					*MS^2^(* *[M+Na]^+^): [M+Na-C_5_H_8_O_2_]^+^*	*351*
55	81.9	Khellactone diacyl ester ^2^ R1–C_5_H_9_O_2_ ^3^ R2–C_5_H_9_O_2_ ^3^	322	C_24_H_30_O_7_ (430)	[2M+Na]^+^	883
					[M+Na]^+^	453
					[M+NH_4_]^+^	448
					[M+H-C_5_H_10_O_2_]^+^	329
56	82.0	Vaginidiol diacyl ester ^2^ R1–C_5_H_9_O_2_ ^3^ R2–C_5_H_9_O_2_ ^3^	322	C_24_H_30_O_7_ (430)	[2M+Na]^+^	883
					[M+Na]^+^	453
					[M+H-C_5_H_8_O_2_-C_5_H_8_O]^+^	245
					[M+H-C_5_H_8_O_2_-C_5_H_10_O_2_]^+^	227
					*MS^2^(* *[M+Na]^+^): [M+Na-C_5_H_8_O_2_]^+^*	*351*
57	82.3	Khellactone diacyl ester ^2^ R1–C_5_H_9_O_2_ ^3^ R2–C_5_H_9_O_2_ ^3^	322	C_24_H_30_O_7_ (430)	[2M+Na]^+^	883
					[M+Na]^+^	453
					[M+NH_4_]^+^	448
					[M+H-C_5_H_10_O_2_]^+^	329

^1^ Identified component and its possible isomers. ^2^ Compound identification was based on the interpretation of UV and MS spectral data and comparison with data from the literature [22,33,34,35,36,37,38]. ^3^ Functional group options are presented in Appendix A. ^4^ Compound identification was based on comparison with the reference standard.

## Data Availability

Data are contained within the article and its Appendix A.

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
