# Peer review of "The Different Composition of Coumarins and Antibacterial Activity of Phlojodicarpus sibiricus and Phlojodicarpus villosus Root Extracts"

_plants, 2024, doi:10.3390/plants13050601_

Round 1

Reviewer 1 Report (Previous Reviewer 1)

Comments and Suggestions for Authors

None

Comments on the Quality of English Language

None

Author Response

Thank you for the comment! We have given the text to the MDPI English editing service

Reviewer 2 Report (Previous Reviewer 2)

Comments and Suggestions for Authors

Dear Authors,

The manuscript entitled “The different composition of coumarins and antibacterial 2 activity of P. sibiricus and P. villosus root extracts” has been revised with great effort and does not need any further corrections.

Author Response

Thank you for your appreciation!

Reviewer 3 Report (Previous Reviewer 3)

Comments and Suggestions for Authors

1. Unfortunately, I was unable to find a response letter regarding responding to the reviewer's comments item by item.

2. When a Latin name appears in the title, the full name should be used.

3. The logic of the abstract is not coherent enough. The combination of component testing and antibacterial activity was not well integrated.

4. There are too many keywords.

5. The outermost layer of Figure 4 should not have a border

6. In the discussion section, it is unnecessary to provide a review of the synthesis pathway and key enzymes of coumarins, as the author did not have any relevant experiments or results in the main text.

7. As the author mentioned in the introduction, the main purpose of this study and its main component (coumarin) is to develop vasodilator drugs. Therefore, the greatest value of this article is to compare the differences in the composition of different plant species, verify their related activities, and provide valuable target compounds for the development of clinical cardiovascular disease treatment drugs. However, the current experiments on antibacterial properties are far from the theme of this article.

8. Regarding the development of antibacterial drugs, simple inhibition experiments on model strains are not enough. The author should conduct targeted inhibition studies based on their research objectives (human disease or plant disease) to demonstrate the potential of this medicinal plant to be developed into an antibacterial agent. Importantly, the inhibition experiments on liquid media are insufficient and inaccurate, and more related supplementary experiments such as solid media are needed.

9. The current workload (HPLC-MS analysis and antibacterial experiments on liquid media) is not sufficient to meet the minimum requirements published in Plants.

10. The expression of English language needs to be greatly improved, and the current situation makes me lack confidence in this article.

Comments on the Quality of English Language

The expression of English language needs to be greatly improved, and the current situation makes me lack confidence in this article.

Author Response

We are grateful to the Reviewer for valuable suggestions and comments that we hope improved our manuscript.

  1. Unfortunately, I was unable to find a response letter regarding responding to the reviewer's comments item by item.

Answer: We are truly sorry that we did not upload the modified text file correctly. We have uploaded it again, with the corrected areas highlighted.

  1. When a Latin name appears in the title, the full name should be used.

Answer: Corrected

  1. The logic of the abstract is not coherent enough. The combination of component testing and antibacterial activity was not well integrated.

Answer: Agree with you, rewritten

  1. There are too many keywords.

Answer: some keywords have been removed or, if possible, could you please delete the words that you think are unnecessary?

  1. The outermost layer of Figure 4 should not have a border

Answer: Revised

  1. In the discussion section, it is unnecessary to provide a review of the synthesis pathway and key enzymes of coumarins, as the author did not have any relevant experiments or results in the main text.

Answer: Shortened this part in the discussion. The discussion provides the biosynthesis pathway and key enzymes of coumarins to accurately reflect the key point at which the synthesis of specifically angular forms of coumarins occurs: angular pyranocoumarins or angular furanocoumarins, to emphasize the importance of this stage. In addition, we are planning to continue this work and detect the key enzymes of this stage in Phlojodicarpus sp.

  1. As the author mentioned in the introduction, the main purpose of this study and its main component (coumarin) is to develop vasodilator drugs. Therefore, the greatest value of this article is to compare the differences in the composition of different plant species, verify their related activities, and provide valuable target compounds for the development of clinical cardiovascular disease treatment However, the current experiments on antibacterial properties are far from the theme of this article.

Answer: Due to the fact that coumarins (in particular, pyranocoumarins and furanocoumarins) have a very wide range of biological activities, the goal of our work was not limited to just finding a replacement for drugs with a proven effect (vasodilator drugs). We tried to study the extracts more carefully. In addition, we received unexpected results: furanocoumarins were found. In this regard, it is very promising to consider other biological activities of these substances based on literature data. Is there a difference in the antibacterial effect of extracts with a predominance of angular forms of substances (furano- and pyranocoumarins)? We also tried to satisfy the requirements of other reviewers.

  1. Regarding the development of antibacterial drugs, simple inhibition experiments on model strains are not enough. The author should conduct targeted inhibition studies based on their research objectives (human disease or plant disease) to demonstrate the potential of this medicinal plant to be developed into an antibacterial agent. Importantly, the inhibition experiments on liquid media are insufficient and inaccurate, and more related supplementary experiments such as solid media are needed.

Answer: Undoubtedly, research will be continued and expanded. We used standard experiments on model objects to understand the directions for further research because furanocoumarins were identified for P. villosus for the first time. At this stage of work, we do not claim to have created/discovered a new antibacterial agent.

  1. The current workload (HPLC-MS analysis and antibacterial experiments on liquid media) is not sufficient to meet the minimum requirements published in Plants.

Answer:

Most of the data presented in the study was obtained for the first time. The presence of furanocoumarins in P. villosus was proven for the first time, and the antibacterial activity of these extracts was obtained for the first time too. Therefore, we would like to publish these data in this journal. And a very interesting fact is that in difficult climatic conditions, angular forms of substances predominate but have different mass spectrometric behavior (furano- and pyranocoumarins). Moreover, the same species at the extreme points of its range has one type of different group of substances.

  1. The expression of English language needs to be greatly improved, and the current situation makes me lack confidence in this article.

Answer:

Perhaps, not being native speakers, we cannot fully convey our thoughts to the reader. The whole text has been checked by English language editing by MDPI. We can provide absolutely all raw data from the devices and their statistical processing. It will take time to reach a higher level of English, we will use additional resources. We will work harder.

Reviewer 4 Report (Previous Reviewer 4)

Comments and Suggestions for Authors

I believe that the manuscript was properly supplemented and modified for publication in Plants.

Author Response

Thank you for your appreciation!

Round 2

Reviewer 3 Report (Previous Reviewer 3)

Comments and Suggestions for Authors

The author has revised it one by one, and the manuscript is recommended for publication.

This manuscript is a resubmission of an earlier submission. The following is a list of the peer review reports and author responses from that submission.

Round 1

Reviewer 1 Report

Comments and Suggestions for Authors

1) Correct the scientific name of the plant.

2) Make the Figure 1 more clearer.

3) Give a brief explanation for Figure 4 caption

4) The other important result of the work is the fairly high content of hydroxycouma- 283 rins in the studied species, P. sibiricus – 13-17 mg/g and P. villosus – 31 mg/g. For P. vil- 284 losus, 30% of all coumarins are hydroxycoumarins, which makes this species a promising 285 source of umbelliferone derivatives. On what basis, this statement is given?

5) The supernatant was filtered and the resi- 339 due was re-extracted once more in the same manner. Make it more precise statement.

6)  Quantification of all identified compounds was 367 performed by peak areas determined at 320 nm using the external calibration method 368 with authentic standards of visnadine and peucenidin (previously isolated from Ferulopsis hystrix). Which method was used? 

7) The mobile phase consisted of a gradient elution of 0.1 % aqueous acetic acid (A) 351 and acetonitrile (B). Verify the statement. 

Comments on the Quality of English Language

Minor grammatical and spell errors. Correct them

Reviewer 2 Report

Comments and Suggestions for Authors

Dear Authors, 

The manuscript entitled “The different coumarins composition from two species Phlojodicarpus” has not been drafted properly and the authors did not make it clear why their work is significant. Also, I strongly believe that the authors should revise the introduction. The changes that need to be addressed are listed below.

1.     The title of the manuscript could be revised since it is not descriptive. The authors have mentioned in the text that they have identified coumarin composition from two species of Phlojodicarpus. I would highly recommend that the authors address both species name in the title to make it more descriptive.

2.     The abstract is written well however, in my opinion, the authors can mention the plant name, and the source of secondary metabolites at the very first for better understanding.

3.     The introduction has not been written properly and needs to be revised. The authors have identified three different coumarins however, they have only mentioned pyranocoumarins. I would suggest here to mention other coumarins also. Moreover, the compound identified from P. sibiricus possess some biological activities, and it is written incomplete and don't provide a clear impression to the readers.

4.     The authors are advised to maintain the text alignment uniformity in the entire manuscript.

5.     English language though good does require improvement in sentence construction like on page no.1 line no.31, page no.2 line no.39-40, 46.

6.     The conclusion part is missing in the manuscript. Authors are suggested to include a proper conclusion related to their study.

7.     The authors are highly advised to maintain the text alignment uniformity about plant species names in the entire manuscript. 

8.     In table.1 author should confirm what are diagnostic ions?

9.     The authors has mentioned “Decursin” in discussion. How this is related with the study since authors does not discussed about decursin in entire manuscript.

10. Author should highlight the limitation of their study along with the future scope of the current work.

Comments on the Quality of English Language

May be improved for acceptability.

Reviewer 3 Report

Comments and Suggestions for Authors

The authors compared different coumarins composition from two species Phlojodicarpus, which provided a new drug source for coumarin compounds, which has a certain scientific significance. Although the authors discussed many of the pharmacological activities of these coumarins. However, I’m more concerned about the biological activity of these substances and their value to human health. Therefore, I think it is necessary to explore further.

Comments on the Quality of English Language

It is noted that your manuscript needs careful editing by someone with expertise in technical English editing paying particular attention to English grammar, spelling, and sentence structure so that the goals and results of the study are clear to the reader.

Reviewer 4 Report

Comments and Suggestions for Authors

The manuscript entitled "The different coumarins composition from two species Phlojodicarpus" covers HPLC/MS analyzes of P. sibiricans and P. villosus extracts.

Coumarins exhibit several pharmacological effects including anticoagulant, antimicrobial, anti-inflammatory, neuroprotective, antidiabetic, anticonvulsant and antiproliferative. Therefore, I think that for the publication in Plants it would be necessary to add some analyzes of the biological activity of the extracts, which would then also give a better application value to the study.